# History of Neonatal Screening of Congenital Hypothyroidism in Portugal

**DOI:** 10.3390/ijns10010016

**Published:** 2024-02-20

**Authors:** Maria José Costeira, Patrício Costa, Susana Roque, Ivone Carvalho, Laura Vilarinho, Joana Almeida Palha

**Affiliations:** 1Life and Health Sciences Research Institute (ICVS), School of Medicine, University of Minho, 4710-057 Braga, Portugal; costeiramj@gmail.com (M.J.C.); pcosta@med.uminho.pt (P.C.); 2ICVS/3B’s—PT Government Associate Laboratory, Braga/Guimarães, 4710-057 Braga, Portugal; 3Department of Neonatology, Unidade Local de Saúde do Alto Ave Rua dos Cutileiros, 4835-044 Guimarães, Portugal; 4Neonatal Screening, Metabolism & Genetics—National Institute of Health Dr. Ricardo Jorge, 4000-053 Porto, Portugal; ivone.carvalho@insa.min-saude.pt (I.C.); laura.vilarinho@insa.min-saude.pt (L.V.)

**Keywords:** neonatal screening, congenital hypothyroidism, thyroid hormones, iodine

## Abstract

Congenital hypothyroidism (CH) leads to growth and development delays and is preventable with early treatment. Neonatal screening for CH was initiated in Portugal in 1981. This study examines the history of CH screening in the country. Data were obtained from annual reports and from the national database of neonatal screening laboratory. The CH screening strategy primarily relies on the thyroid-stimulating hormone (TSH), followed by total thyroxine measurement as the second tier for confirmation. The TSH cutoff started at 90 mIU/L, decreasing to the actual 10 mIU/L. The coverage of the screening program has increased rapidly; although voluntary, it reached about 90% in 6 years and became universal in 10 years. Guideline and cutoff updates led to the identification of over 200 additional cases, resulting in specific retesting protocols for preterm and very-low-birth-weight babies. The actual decision tree considers CH when TSH levels are above 40 mIU/L. Data from the CH screening also provide an indication of the iodine status of the population, which is presently indicative of iodine insufficiency. The Portuguese neonatal screening for CH is a history of success. It has rapidly and continuously adapted to changes in knowledge and has become a universal voluntary practice within a few years.

## 1. Introduction

Neonatal metabolic screening began in the early 1960s, when Prof. Robert Guthrie demonstrated that blood collected in a filter paper card (“dried blood spots”) could be used to assess biochemical parameters [1]. Later, in 1972, Prof. Jean Dussault boosted congenital hypothyroidism (CH) screening by measuring blood levels of thyroxine (T4) in paper cards, justifiably named Guthrie cards [2].

In Portugal, the national neonatal screening program started in 1979, with the screening of phenylketonuria. In 1981, the screening of (permanent) CH was added, and from then on, the number of evaluated conditions continuously expanded until reaching the present 26.

CH is the most common cause of neonatal thyroid disease. If not treated in time, it can lead to neurodevelopmental and intellectual disabilities. CH can be classified as permanent (requiring lifelong treatment) or transient (requiring treatment for a defined period, most commonly the first three years of life). Transient CH can be caused by secondary factors such as iodine deficiency or excess and maternal and neonatal conditions (e.g., transplacental passage of maternal antibodies or drugs, prematurity, non-thyroid disease, neonatal use of dopamine), among others [3,4,5,6,7]. Permanent CH is caused by a morphological abnormality or dysfunction of the thyroid, which results in impaired thyroid hormone synthesis. Another rarer case of permanent hypothyroidism, called central CH, is caused by abnormal function of the hypophysis, leading to decreased levels of thyroid-stimulating hormone (TSH) and thyroid hormones.

The main purpose of CH-screening programs is to detect all forms of primary CH, whether mild or severe. Most middle- and high-income countries have implemented national CH screening for early detection and treatment (measurement of TSH and/or total thyroxine (TT4) in a heel-prick blood sample collected on filter paper). Consensus updates recommend retesting of newborns at high risk of CH, namely, preterm babies, very-low-birth-weight (VLBW) babies, twins, and sick babies [6,7,8]. A recent metanalysis reported a pooled worldwide prevalence of 1 in 1901 in the decade 2011/2020 [9].

In pregnancy, until the 20th week of gestation, the fetus completely depends on the maternal thyroid hormone supply, relying on proper placental function and on the mother’s thyroid status and iodine sufficiency. The fetal thyroid gland only starts to produce hormones by mid-gestation. As such, the fetus relies on the maternal supply until delivery. Thyroid hormones are crucial for normal cellular homeostasis and brain development. Thyroxine (T4), the main hormone synthesized and secreted by the thyroid gland, is converted in tissues into the biologically active hormone triiodothyronine (T3). The hypothalamic–pituitary–thyroid axis responds to changes in the concentration of free thyroid hormones, which modulate the amount of pituitary-released TSH to stimulate the thyroid gland [6,10,11,12].

For neonatal screening, the timing of sample collection and retesting considers the thyroid’s physiology. At birth, there is a spike in TSH that results in increased T4, which lasts two days. These physiological responses justify sampling for neonatal screening on days 3–6 of life, which is also the adequate time to screen for other metabolic diseases. In premature (especially those less than 32 weeks old), with very low birth weight (VLBW), iodine-deficient, iodine-overload infants (caused by disinfecting agents), and babies with other medical conditions (for example, certain drugs, maternal thyroid disorders), the hormone increase is impaired or delayed, which justifies retesting. All newborns detected by screening should be referred for diagnostic confirmation, immediate treatment, and adequate follow-up.

Despite the success and cost-efficiency of the screening program, many low-income countries have yet to achieve universal nationwide coverage: according to the consensus published in 2021 and 2023, 70% of babies around the world still have no access to metabolic screening, and most of them live in iodine-deficient areas, which increases the risk of neurodevelopment impairment in the most vulnerable period of life [6,7,13].

The present study intends to share the history of neonatal screening of congenital hypothyroidism in Portugal and to discuss its determinants and challenges throughout time.

## 2. Materials and Methods

The newborn screening program is conducted by the designated National Program of Neonatal Screening, previously named the National Program of Early Diagnosis [14].

The data used to perform this work were mainly gathered from annual reports from 1981 to 2022, and from the database of the laboratory in charge of the screening. Missing information was retrieved upon consultation with the direction of the program.

The following information was collected: number of newborns per year, number of screened babies, date of blood collection, number of positive cases (permanent and transient), screening cutoffs and approach, recall rate, day of communication of positive cases, and global results of TSH and TT4 measurements.

In the CH screening program, the TSH test strategy is used as the primary test, and the measurement of TT4 is used as a secondary test to confirm the diagnosis; both use Auto-DELFIA methods.

## 3. Results

Table 1 summarizes the indicators of the neonatal screening program for CH in each decade from the beginning to 2021 (last complete year).

Figure 1 shows how the screening program’s coverage rapidly grew, reaching about 90% in 6 years and becoming universal within 10 years.

The screening methodology followed the best practices originating from evidence at each given time, and, therefore, was adapted throughout time. Table 2 depicts the dates on which new policies were implemented. Sample collection ages ranged from the 5th–10th to the actual 3th–6th day; the screening methodology varied from using solely TSH with varied cutoff values to the inclusion of TT4 measurement as a 2nd tier test and retesting of more vulnerable babies.

TSH cutoff values for recall and the screening algorithm decreased from 90 mIU/L to the present 10 mIU/L. Since 2006, TSH levels in the range of 10–20 mIU/L have implied an additional measurement of TT4 (whenever TT4 < 9,5 µ/dL, a new sample is requested for repetition). In the case of preterm babies less than 32 weeks and/or VLBW babies (birth weight < 1500 g), by default, TSH and TT4 have been measured a second time (at two weeks) since 2009; a third time (at four weeks) since 2014; a fourth time (at 36 weeks or at the time of hospital discharge) since 2020; and a fifth time (32 weeks, 36 weeks, or by the time of hospital discharge) after 2021. Interestingly, in some time periods, there were various cutoffs for the diagnosis to be considered normal, which had implications regarding the balance between sensitivity and specificity; this stabilized from 2006 onwards.

Lowering TSH cutoff values and adding a 2nd tier was intended to increase the detection rate. For each change in methodology, one can evaluate the number of cases that would have been missed if using the previous one. As shown in Figure 2, updating the guidelines throughout time allowed for the identification of 200 babies that would otherwise have had missed diagnoses.

In the 42 years under analysis (1981–2022), 4.185.861 newborns were tested, of which 1502 babies were identified as having permanent CH and were referred for replacement treatment with levothyroxine. The rate of positive cases increased by 24% from 1990 (considered the beginning of universal coverage) to 2021. In the same period, more than 2814 other babies were found to have transient CH.

In the period of 2012 (the first annual report with information regarding sex) to 2021, cases of CH were found to be more frequent in females, with a distribution of 56% in female vs. 44% in male newborns (χ2(1) = 8.81, *p*= 0.003; OR = 1.36).

The time from sample collection to communication of abnormal results has decreased to the present 9.3 days, which allows treatment to be started by ten days of life (Figure 3).

Screening CH in preterm and sick babies is challenging, since their thyroids are in diverse stages of development. For this reason, a second-tier test and additional retesting are necessary. Table 3 shows that half of the babies with CH (information available between 2014 and 2021) required 2–3 measurements in order to achieve a diagnosis. Of these, 10–40% (2016–2021) were premature, with more than half displaying thyroid dysfunction typical of prematurity. In a few preterm babies, TT4 levels were always within the normal reference range.

In order to start treatment as soon as possible, all intermediate steps between sample collection and arrival at the lab must be optimized. In recent years, due to reorganizations of Primary Care Units and the Postal Service, the transit times of samples have increased: in 2010, 22% of the total samples arrived in the lab one day after collection, and in 2021, those values fell to 12%; still, most of the samples arrive within the optimal time period.

The etiology of CH is not always identified in reports, and is only mentioned in three of them: based on imaging studies, most cases are due to thyroid dysgenesis/agenesis (74% in 2000, 64% in 2005, and 89% in 2006).

Universal neonatal screening provides additional information, including an indication of the population’s iodine status. Iodine insufficiency is suggested when over 3% of the newborns in the neonatal screening display TSH > 5 mIU/L. Presently, Portugal seems to be on the edge (Table 4) regarding that condition.

The population’s adhesion and trust in the screening program can be observed by the number of parents (75%) voluntarily taking their babies to Public Primary Care Units to perform the screening; the other samples are collected while still in the hospital and in private clinics. Despite the growing number of deliveries in private hospitals and clinics (14% in 2013 and 18% in 2021), families voluntarily go to public health centers (in 2021, only 12% of samples were collected in private units). It should be highlighted that both public and private units comply with the expected guidelines of the neonatal screening program.

Parents have been able to access the screening results online on the institutional website since 2003–2004. Parents’ access to the webpage is increasing, reaching 51% in 2021. Parents can express their satisfaction with the program on the same website, which has been over 75% (very satisfied) for the last seven years.

## 4. Discussion

CH fulfills all criteria to be mass-screened in the neonatal period. According to the principles of Wilson and Jungner [15], it is an important health issue with an easy, effective, safe, and affordable treatment. It increases population health, with well demonstrated cost-effective and social benefits, eliminating cases of intellectual disability due to untreated CH [6,7].

Portugal, like many other European countries, Japan, and some states of United States of America, chose to employ TSH as the first tier for diagnosis at the beginning. Countries like Australia, Israel, New Zealand, and some states in the United States of America instead chose the T4 test strategy [5,6,7,8]. The advantages and disadvantages of the various options have been widely discussed, and differences in the reported birth prevalence and recall rates of CH, delayed TSH rise, and subclinical hypothyroidism diagnosis have been justified [6,13,16]. Briefly, the screening based on TSH levels better identifies primary CH (even mild cases), while that based on T4 performs better for central CH and when TSH elevation is delayed. However, it is more difficult to assess preterm babies and mild cases of primary CH. When both TSH and T4 are used, this allows for the detection of all forms of hypothyroidism, despite being more expensive [6,7,16].

Worldwide, the main goal of screening programs has evolved to include the identification of milder forms of CH which would not otherwise have been treated.

The recall rate for CH was 0.08% in 2021, which is slightly below that reported in the literature (most probably because in Portugal, sample collection always occurs after two days of life), and that of premature babies was 1.3% (2020), similar to other published observations [17,18,19]. Thyroidal dysfunction typical of prematurity was found more frequently than reported, since, by default, all preterm babies under 32 weeks of gestational age are retested. In the last seven years, delayed hyperthyrotropinemia was seen in 64% of severe cases of permanent CH; with time and changes in cutoff values, the metabolic screen evolved to detect milder forms of the disease. The measurement methods also changed from the initial radioactive RIA and IRMA methods, which used I^125^, to non-radioactive DELFIA approaches, which are safer for laboratory personnel.

The effectiveness of early detection and timely treatment can ultimately determine the program’s quality. As such, the Portuguese screening program reached its aims: it attained 100% coverage ten years after launching, despite not being mandatory. It is worth mentioning that it reached 97% in 1993, a time when 5% of babies in Portugal were still home-delivered (without any health assistance) and the mothers took their babies to health units to perform metabolic screening.

The communication pathway was well established, and treatment time quickly dropped to 10 days in 2006, remaining inside the advised two-week window since then [6,7].

Improving the detection rate, including milder forms of CH, and decreasing the number of potential missing cases led to changes in practice over time: earlier blood collection to speed up treatment, decreased TSH cutoffs levels, and particular attention to vulnerable babies (premature, VLBW, twins and sick babies) [10,18,20,21,22,23]. All these measures were counterbalanced with the expected spur of recall rate and the secondary (but not negligible) worries of families’ anxiety with false positive results [6,7,24].

In Portugal, TSH cutoffs decreased from 90 mIU/L in 1981 to the actual 10 mIU/L, as the 2nd tier test (TT4) was included from 1989 on for those above the TSH threshold. Following the best available evidence, the changes in the cutoff over time allowed the identification of more than 200 babies with CH. It also allowed the identification of hypothyroxinemia of prematurity in 59% of preterm infants less than 32 weeks and/or less than 1500 g who were retested. These routine and standardized preterm rescreening practices have been in place since 2016. They represent an example of effective communication between clinicians and the laboratory, as well as adherence by healthcare providers [6,7,16]. Other groups may also require particular attention, and research is needed in order to better understand the influence of other risk factors on newborn thyroid dysfunction (e.g., being small for gestational age, twins, acute illness, medications, admission to intensive care units).

Over time, a rise has been observed in the reported incidence of CH, which has also been observed worldwide [6,7,25,26,27]. This is attributed to the lowering TSH cutoffs that allow for the diagnosis of biochemically milder forms of CH; higher prematurity rates; multiple pregnancies; and environmental (namely, iodine deficiency), genetic, and ethnic factors [6,7,17,20,22,28,29,30]. Notably, a change in interpretation has also been established for premature babies, many of which were initially included as having transient hypothyroidism due to immaturity of the thyroid axis. From 2009 onwards, preterm babies have begun to be retested, and, therefore, reported as normal in most cases.

Of interest, we, as some others, observed a slightly higher incidence of CH in females, which might be attributed to genetic factors [4,16,27,31,32,33,34]. This is not a consensus, and may possibly be ethnically influenced [35,36,37].

Portugal is a small country (<100.000 km^2^), with most of the population living in coastal areas. We found no geographic differences in the identified rates of CH that could be attributed to the distance from the sea. Despite being considered an iodine-sufficient country [37,38], as measured by the mean urinary iodine in schoolchildren, one-third of children still have iodine levels lower than desired. The data presented here on the percentage of newborns with TSH > 5 mIU/L further denote a state of iodine insufficiency [38]. Of relevance, women of childbearing age and pregnant women have been reported to be iodine-deficient [39,40], which led the local authorities in 2013 to recommend supplementation with iodine during preconception, pregnancy, and lactation [41].

It is worth mentioning the engagement of the parents in the screening process. The current procedures include issuing a report that can be retrieved from a web-based page where the results are available. Still, since abnormal results are communicated directly by phone, less than 50% of the parents consult the report. Interestingly, parents can provide suggestions that are taken, as is the case for having the collection procedures written on the reverse of the Guthrie cards, which was a father’s idea.

Despite the positive evolution, the screening program faces some challenges. One difficulty is the delay between blood collection and sample arrival in the laboratory due to postal postponing: only 12% of the samples arrive in the laboratory the day after collection, since many health centers are mailing the cards only twice a week or even charging parents to post the Guthrie cards, which delays diagnosis—this problem persists, despite the alerts given.

A final word for the organization of the CH screening program: The national newborn screening in Portugal is centralized in one national laboratory (this has been the case almost from the beginning) responsible for deciding the testing methodologies and communicating with the health professionals, as well as providing care to newborns (from doctors and nurses to health managers) and families. By default, all babies diagnosed with CH are referred to eight centers that follow the standard protocols and guidelines to confirm the diagnosis, start treatment, study the etiology, and follow up on the children.

Neonatal screening programs for CH are widely considered to be an achievement of public health, and Portugal is an example of that success. In spite of this, the program faces challenges, such as transit times for the samples, determining the best approach for at-risk populations besides premature babies, and elucidating some issues such as sex and genetic determinants.

## Figures and Tables

**Figure 1 IJNS-10-00016-f001:**
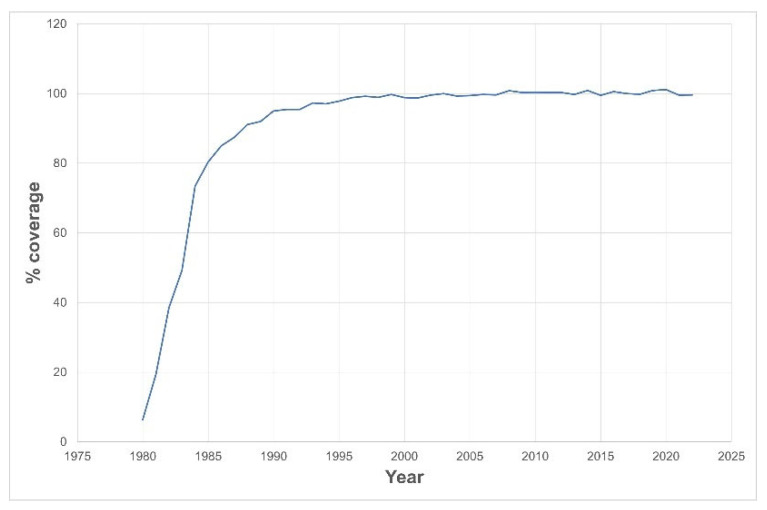
Neonatal screening coverage rate since launch of the screening program.

**Figure 2 IJNS-10-00016-f002:**
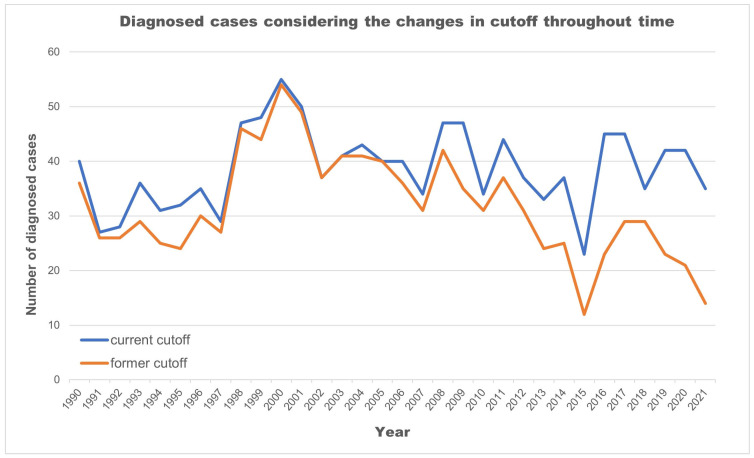
Number of cases further diagnosed given the cutoff updates (1981–2021).

**Figure 3 IJNS-10-00016-f003:**
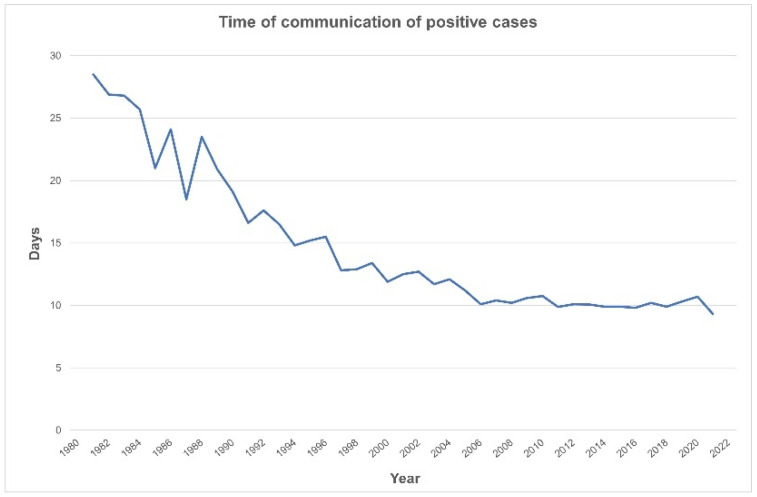
Decreasing time to notification of a positive screen (1981–2021).

**Table 1 IJNS-10-00016-t001:** Summary of the main indicators of the screening program in specified years.

	Year					
Program indicator	1981 ^#^	1990	2000	2010	2020	2021 ^##^
Number of births	152,102	116,383	120,071	101,381	84,426	79,582
Number of screened newborns	29,323	110,607	118,554	101,773	85,456	79,217
Coverage rate (%) ^&^	19.3	95	98.8	100.4	101.1	99.5
Cumulative incidence	1/4300 *	1/3900	1/3367	1/3045	1/2841	1/2827
Beginning of treatment (days)	30 *	19.1	11.9	10.4	10.9	9.3
% Recall CH		0.09	0.27	0.33	0.07	0.08
% Recall preterms			0.0029 **	0.1828	1.326	NA

^#^: First year of available data. ^##^ Included as the latest available data. ^&^: Newborn samples collected in the last days of December are considered as belonging to the following year, which implies small fluctuations in the %. * Reported from 1982 onwards; **: reported from 2006 (when 2nd sample collection began) onwards.

**Table 2 IJNS-10-00016-t002:** Evolution of screening methodology throughout time.

Year	Sample Collection (Day)	Decision Tree		Retesting Preterm Babies	
		TSH Cutoff (mIU/L)	Procedure	Criteria	Procedure
1981–1988	5–10	<90–80–50 ^#^	normal	NA	NA
		>90–80–50 ^#^	referral		
1989–1996	4–7	<30	normal		
		>30	referral; use of TT4 as 2nd tier		
1997–2002	4–7	<20	normal		
		20–40	2nd sample		
		>40	referral; use of TT4 as 2nd tier		
2003–2005	3–6	<20	normal		
		20–40	2nd sample		
		>40	referral; use of TT4 as 2nd tier		
2006–2008	3–6	<10	normal		
		10–20	2nd tier- if TT4 < 6.5 µg/dL: 2nd sample		
		>20	referral; use of TT4 as 2nd tier		
2009–2014	3–6	<10	normal	PT < 30 wGA and/or BW < 1500 g	Retesting: 14–15 days
		10–20	2nd tier- if TT4 < 6.5 µg/dL: 2nd sample		
		20–40	2nd sample		
		>40	referral; use of TT4 as 2nd tier		
2015	3–6	<10	normal	PT < 30 wGA and/or BW < 1500 g	Retesting: 14–15 and 28–30 days
		10–20	2nd tier if TT4 < 6.5 µg/dL: 2nd sample		
		20–40	2nd sample		
		>40	referral; use of TT4 as 2nd tier		
2016–2017	3–6	<10	normal		
		10–20	2nd tier if TT4 < 8.5 µg/dL: 2nd sample		
		20–40	2nd sample		
		>40	referral; use of TT4 as 2nd tier		
2018–2020	3–6	<10	normal		
		10–20	2nd tier if TT4 < 9.5 µg/dL: 2nd sample if TSH >10 mIU/L: refer		
		20–40	2nd sample if TSH > 10 mIU/L: refer		
		>40	referral; use of TT4 as 2nd tier		
2021–2022	3–6	<10	normal	PT < 27 wGA	Retesting: 14–15, 28–30 days, 32 and 36 weeks
		10–20	2nd tier if TT4 < 9.5 µg/dL: 2nd sampleif TSH >10 mIU/L: refer	PT 27–30 wGA PT 30–32 wGA	Retesting: 14–15, 28–30 days and 36 weeks
		20–40	2nd sample if TSH >10 mIU/L: refer		
		>40	referral; use of TT4 as 2nd tier		

PT: preterm babies; GA: gestational age; wGA: weeks of gestational age. ^#^: In this time range, various cut-offs were considered. What is reported is the overall decision tree for the correspondent period of time.

**Table 3 IJNS-10-00016-t003:** CH among preterm babies (information available in annual reports 2014–2021).

	2014	2015	2016	2017	2018	2019	2020	2021
Number of CH cases	37	23	45	45	35	42	42	35
Number (and %) of CH cases who required 2–3 measurements for diagnosis	12 (32%)	6 (26%)	22 (49%)	24 (53%)	24 (69%)	17 (40%)	22 (52%)	22 (63%)
Number of PTs of the ones (with CH) who had 2–3 samples	NA	NA	2	5	6	4	8	9
Number of PTs with TSH < 10 mU/L in 1st sample	NA	NA	2	5	0	3	7	8
Number of PTs with TSH < 10 mU/L in 2nd–3rd samples	NA	NA	2	2	0	2	5	6
Number of PTs with normal TT4 at all times	NA	NA	2	1	0	1	3	2

NA—not applicable.

**Table 4 IJNS-10-00016-t004:** Screened babies with TSH > 5 mIU/L (%) between 2013 and 2022.

Year	Screened Babies with TSH > 5 mIU/L (%)
2013	2.5
2014	2.9
2015	2.7
2016	3.0
2017	3.6
2018	3.0
2019	2.9
2020	2.9
2021	3.4
2022	3.3

## Data Availability

The data will be made available upon request to the corresponding author.

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
