# Peer review of "History of Neonatal Screening of Congenital Hypothyroidism in Portugal"

_2409-515X, 2024, doi:10.3390/ijns10010016_

Round 1

Reviewer 1 Report

Comments and Suggestions for Authors

 History of neonatal screening of Congenital Hypothyroidism in Portugal 

This is an important paper as it reviews decades of data. It is an excellent exercise to gather lessons  on implementation issues. Studying the CH program is a good start and can serve as a template for reviewing the other disorders covered in the newborn screening pane.

1.    For the introduction, 

Include the burden of CH. There are good articles to consider.

You already cited Rose et al "Congenital Hypothyroidism: Screening and Management." Pediatrics 151, no. 1 (2022) which has mention of the worldwide incidence.

Another Article To Consider Is Current Status Of Newborn Screening Worldwide: 2015 By Therrell Et Al, Seminars In Perinatology 39(2015)171–187 

2. Consider the review article ‘Thyroid Function across the Lifespan: Do Age-Related Changes Matter?’ Endocrinology and Metabolism 2022;37:208-219 https://doi.org/10.3803/EnM.2022.1463 pISSN 2093-596X · eISSN 2093-5978

There are 2  interesting sections for your consideration:   environmental influences and thyroid function in utero

3. Results - General comment for all tables and figures. Please review the titles for better description  as well as labels of column/row labels.

4.     Results - Table 1. Please clarify. 

Number of births  and Number of screened newborns:  Is this for the year 1981 or for the decade?

Are you comparing births every 10 years? 

Labels need to be improved.

5.     Results -Table 2 has to be improved.

Column 3 is vague. What does the following  mean?

< 90-80-50 (for and backs)

>90-80-50

<20-40 (40) (for and backs)

20-50 (40)

<15-20 (for and backs)

20-40

>40

Etc

Footnotes may help

6.     Results - Figure 2 Label of Y axis is vague- are you referring to positive screens or positive, diagnosed cases?

Further diagnosed cases – are you referring to the orange shaded part?

You had several changes in cut off. I wonder if a line diagram or a bar graph is a better?

7.     Lines 157-160 in relation to Table 4 needs clarification. Is this saying that these patients have iodine deficiency and these were the TSH Results.

8.     Line 197. Which is correct – six or ten years after launching?

9.     Although this paper is only on screening for CH, are there efforts to reduce the timing of sample collection further since other conditions are also being covered in the panel. There should at least be included in the discussion.

Comments on the Quality of English Language

It is generally acceptable but needs polishing.

Author Response

Reviewer 1

This is an important paper as it reviews decades of data. It is an excellent exercise to gather lessons  on implementation issues. Studying the CH program is a good start and can serve as a template for reviewing the other disorders covered in the newborn screening pane.

Thank you for recognizing the relevance of reporting such type of data. 

  1. For the introduction, Include the burden of CH. There are good articles to consider.

You already cited Rose et al "Congenital Hypothyroidism: Screening and Management." Pediatrics 151, no. 1 (2022) which has mention of the worldwide incidence.

Another Article To Consider Is Current Status Of Newborn Screening Worldwide: 2015 By Therrell Et Al, Seminars In Perinatology 39(2015)171–187 

Thank you for your note. We have now included the information obtained from a recent meta-analysis on the worldwide prevalence (Liu L, He W, Zhu J, Deng K, Tan H, Xiang L, Yuan X, Li Q, Huang M, Guo Y, Yao Y, Li X. Global prevalence of congenital hypothyroidism among neonates from 1969 to 2020: a systematic review and meta-analysis. Eur J Pediatr. 2023 Jul;182(7):2957-2965. doi: 10.1007/s00431-023-04932-2. Epub 2023 Apr 18. PMID: 37071175.) Current version lane 53-54.

  1. Consider the review article ‘Thyroid Function across the Lifespan: Do Age-Related Changes Matter?’ Endocrinology and Metabolism 2022;37:208-219 https://doi.org/10.3803/EnM.2022.1463 pISSN 2093-596X · eISSN 2093-5978

There are 2  interesting sections for your consideration: environmental influences and thyroid function in utero

We are thankful to the reviewer. We do agree that the aspects on environmental challenges (iodine, as reported in the article, and also endocrine disruptors, many of which depend on the type of diet) and thyroid function in utero are relevant.

In agreement, above 3% TSH>5mIU/L in the neonatal screening, supports iodine deficiency. We highlight this aspect in the article, since this further supports other date also suggestive of iodine deficiency in the Portuguese population.

  1. Results - General comment for all tables and figures. Please review the titles for better description  as well as labels of column/row labels.

We have now revised and altered for better understanding.

  1. Results - Table 1. Please clarify. Number of births  and Number of screened newborns:  Is this for the year 1981 or for the decade? Are you comparing births every 10 years? Labels need to be improved.

We have now clarified. The data on the table refers to the first year in each decade (1990, 2000, 2010, 2020), except for 1981 (the first year of available data) and 2021 (the latest available data). We have now revised the title of the table and justified the exceptions in the footnote of the table.

  1. Results -Table 2 has to be improved.

Column 3 is vague. What does the following mean? < 90-80-50 (for and backs) >90-80-50

<20-40 (40) (for and backs)       20-50 (40)        <15-20 (for and backs)  20-40

>40      Etc       Footnotes may help

We have now revised. Interestingly, in some time periods, different cut-offs were tried. We have now included this information as a footnote (lane 120-122).  of the table and also in the text (lane 130-132). We have also redone the table and consider that it now reads better. Thank you.

  1. Results - Figure 2 Label of Y axis is vague- are you referring to positive screens or positive, diagnosed cases? Further diagnosed cases – are you referring to the orange shaded part? You had several changes in cut off. I wonder if a line diagram or a bar graph is a better?

Thank you for your comment. Throughout the year, whenever a cut-off has changed, additional cases could have been detected. We have now changed the labels and the format of the graph for better understanding. 

  1. Lines 157-160 in relation to Table 4 needs clarification. Is this saying that these patients have iodine deficiency and these were the TSH Results.

We have now rephrased the sentence into “Universal neonatal screening provides additional information, including an indication of the population's iodine status. Iodine insufficiency is suggested when over 3% of the newborns in the neonatal screening display TSH>5 mIU/L. Portugal seems to be presently on the edge (Table 4) for that condition.” (lines 170-173)

This aspect is also discussed on lanes 255-258.

  1. Line 197. Which is correct – six or ten years after launching?

Thank you for noticing. It has been corrected. It is ten.

  1. Although this paper is only on screening for CH, are there efforts to reduce the timing of sample collection further since other conditions are also being covered in the panel. There should at least be included in the discussion.

The timing for the neonatal screening not only includes the physiological changes considered in the thyroid axis, but also the time needed for feeding, which is required for the diagnosis of other metabolic disorders. A reference is now made to this on lanes 68, when timing of collection is addressed.

Reviewer 2 Report

Comments and Suggestions for Authors

Costeira et al wrote a nice review of the history of newborn screening experience for congenital hypothyroidism in Portugal, spanning more than 30 years.

I think the manuscript will benefit from few minor modifications.

Table 1: in 2010 and 2020 the number of screened newborns was higher than the number of births.

Please explain how that is possible (number of specimens screened instead of newborns?)

Table 2 would be easier to follow if transformed in a figure, something like hierarchy smartart

Line 131-133: the rate of positive cases increased 24% from 1990 (considered….) to 2021.

This sentence is difficult to understand. You should use different metrics, like detection rate, and possible also true positive, false positive and false negative metrics (if possible). It’ll improve the overall quality and significance of the manuscript if you calculate them for both permanent and transient CH and expand this section accordingly.

You talked briefly about iodine insufficiency and pointed out the geography of Portugal. I was wondering if it’d be possible to evince any geographical distribution.

Author Response

Reviewer 2

Costeira et al wrote a nice review of the history of newborn screening experience for congenital hypothyroidism in Portugal, spanning more than 30 years.

I think the manuscript will benefit from few minor modifications.

Thank you for your consideration.

Table 1: in 2010 and 2020 the number of screened newborns was higher than the number of births.

Please explain how that is possible (number of specimens screened instead of newborns?)

Thank you. The newborn samples collected in the last two weeks of December are considered in the following year, which implies small fluctuations of on the %.  We have now included this information as a footnote. (lane 102-103)

Table 2 would be easier to follow if transformed in a figure, something like hierarchy smartart

Thank you, we have remade the table for clarification. We considered your suggestion of using a decision tree map, but it looked more difficult for comparisons to be made.

Line 131-133: the rate of positive cases increased 24% from 1990 (considered….) to 2021.

This sentence is difficult to understand. You should use different metrics, like detection rate, and possible also true positive, false positive and false negative metrics (if possible). It’ll improve the overall quality and significance of the manuscript if you calculate them for both permanent and transient CH and expand this section accordingly.

Thank you for the note, which allows us to further clarify the changes in transient hypothyroidism. We have now clarified in the discussion, lanes 245-248: “Of notice, a change in interpretation has also been given to premature babies, many of which were initially included as having transient hypothyroidism due to immaturity of the thyroid axis. From 2009 onwards, preterm babies started to be retested and therefore, in most cases, reported as normal.

You talked briefly about iodine insufficiency and pointed out the geography of Portugal. I was wondering if it’d be possible to evince any geographical distribution.

We have now included this information in the discussion, as data not shown: “We found no geographic differences in the identified rates of CH that could be attributed to the distance from the sea (data not shown).” (Lanes 253-254)
